# Adult outcomes by parental, school and postcode aggregated income in childhood—A descriptive analysis of the cohorts 1981–1989 in Finland

Aapo Hiilamo[1,2]*, Niilo Luotonen[3,4], Lauri Mäkinen[5,6], Tiina Ristikari[5,7]

**1** Max Planck Institute for Demographic Research, Rostock, Germany, **2** Max Planck – University of Helsinki Center for Social Inequalities in Population Health, Rostock, Germany and Helsinki, Finland, **3** VATT Institute for Economic Research, Helsinki, Finland, **4** Aalto University, Espoo, Finland, **5** Itla Children's foundation, Helsinki, Finland, **6** Social Insurance Institution of Finland, Helsinki, Finland, **7** Väestöliitto, the Family federation of Finland, Helsinki, Finland

* hiilamo@demogr.mpg.de

## Abstract

Monitoring the ways in which childhood socioeconomic environment is associated with adult outcomes is fundamental in terms of informing evidence-based debates. However, we know little about the cohort differences in the association of childhood income with adult outcomes in Finland, a country of low economic inequality and one of the most desegregated educational systems in the OECD countries. We use register data on cohorts born in 1981−89. We divide these cohorts into tenths by their parental, school and postcode-level aggregated income at the age of 15. We focus on five adult outcomes at the age of 30. On average across the cohorts, compared to the highest income groups, the lowest tenth of parental, school and postcode income had, respectively, a 1.8, 1.7 and 1.4-fold risk of death, 3, 1.3 and 0.9-fold risk of low education, 2.5, 1.8, and 1.4-fold risk of long-term unemployment. People with a lower childhood parental or school income were more likely to live with a partner and have children than people from higher income backgrounds. The differences by school and postcode income in the education outcome were small. We observe increasing differences in the education and employment outcomes by parental and, to a lesser extent, school income.

## Introduction

A vast body of research investigates the extent to which childhood socioeconomic environment matters for later life. This research shows that children in low income families have a higher risk of negative health, demographic and economic adult outcomes, net of other individual characteristics [1,2]. Research on neighbourhood effects also demonstrates that adult outcomes relate to the areas in which children

**Data availability statement:** The authors cannot share the data due to privacy laws. Data is provided by Statistics Finland and can be applied via (https://www.tilastokeskus.fi/tup/tutkijapalvelut/aineistot-ja-niiden-toimitusmu-odot_en.html). We provide the aggregated data and code used to produce the aggregated data online: https://osf.io/nghk8. The individual level data was analysed in secured Fiona remote server.

**Funding:** Luotonen gratefully acknowledges funding from the Kone Foundation (202006073). The funders had no role in study design, data collection and analysis, decision to publish, or preparation of the manuscript.

**Competing interests:** The authors have declared that no competing interests exist.

grow up and form relationships; living in a deprived area in childhood is associated with low educational attainment, poorer health and lower income, after adjusting for other factors (For a review of economic outcomes see [3], for health outcomes see [4] and for education outcomes [5]). Moreover, research on 'school effects' finds that the socioeconomic composition of the schools that children attend is linked to their adult outcomes [6,7].

But do we really know the true extent to which childhood socioeconomic environments at these levels are associated with adult outcomes? Previous studies often examine adult outcomes by childhood socioeconomic status (SES) indicators while combining multiple birth cohorts and adjusting for other characteristics. These adjusted characteristics may include individual-level characteristics or other dimensions of SES in childhood or adulthood, such as parental wealth status (see review by [8]) These adjustments, however, make interpretation of their estimates difficult and hide the description of the factual population level patterns. The adjusted factors in these studies may be conceptualised as confounders or as mediating factors [9], for example, as mechanisms through which high income schools produce better outcomes.

There is clearly a descriptive gap in the literature. A strictly descriptive investigation examines population parameters, such as factual population means, their differences and time trends in the population without modelling assumptions and adjusting for other factors. Descriptive work is required in order to monitor the state of social inequalities over time and to inform the general public about the current state of equal opportunities. Descriptive research helps policymakers to determine the need for policy measures (for a more detailed discussion, see [10]).

This study seeks to offer a descriptive account of the ways in which childhood income is associated with adult outcomes in Finland. We focus on childhood income aggregated at three levels – parents, schools and postcodes – and five key adult demographic and socioeconomic outcomes.

From an international perspective, Finland provides an interesting laboratory for this inquiry. The country is often considered to be an exemplar of social equality with low income and wealth inequality, an extensive welfare safety net and one of the most desegregated educational systems in the OECD countries (see overview by [11]). The level of child poverty is one of the lowest in the European Union. The reason for this is progressive taxation and the role of social transfers at the lower end of the income distribution [12]. The Finnish education system has historically produced equitable learning outcomes and is almost completely public and free of charge [13]. The school system has been built around an ethos of providing equal opportunities for further education for all students with different backgrounds [14].

Despite these institutional factors, socioeconomic environment in childhood continues to matter for later life, even in Finland. Parental tertiary education predicts children's tertiary education, parental income predicts children's later income, and so on. [15–19] While the role of parental income and education for later outcomes has been reported to be weaker, in magnitude, in Finland than in comparable countries [20–22], Finland is still unable to provide, never mind guarantee, equal opportunities in terms of adulthood socioeconomic outcomes.

We examine here whether there is any meaningful birth cohort variation in the associations of childhood socioeconomic environment with adult outcomes among those born in the period 1981−1989. These cohorts are unique. They experienced not only two significant economic downturns during their early life course, but also a major societal transformation with the declines in family formation and fertility in the 2010s (see, e.g., [23]). The first economic downturn was the Finnish recession of the early 1990s, leading to a sharp increase in unemployment rates, from circa 3 per cent in 1990 to a peak of almost 17% in 1994 [24]. The recession had a lasting impact on the Finnish society, including increased poverty rates, social inequality and mental health problems. The cohorts investigated here experienced the worst phase of the recession in their pre-school years (cohorts 1986−9), their pre-teen years (cohorts 1982−5), or in their teenage years (cohort 1981). The second economic downturn experienced by these cohorts was when they entered the labour market, namely, the great recession after the financial crisis of 2008. There are debates around whether these recessions and the austerity measures related to them further polarised the opportunities for education and employment by childhood socioeconomic background (e.g., [25]) though little in the way of empirical examination exists in respect of these trends.

The demographic context here is that the country experienced a massive decline in family formation during the period these cohorts entered adulthood. For instance, the number of new marriages declined from 71 per 10 000 adults (30 557 new marriages) in 2011–50 (22 296 new marriages) in 2019 according to Statistics Finland, amounting to a drop of 30 percent [26]. During the same period the total fertility rate declined from 1.83 to 1.35 [27]. It is however debated to what extent economic circumstances are the driving forces behind these trends. Investigating how these trends are patterned according to childhood socioeconomic environment can provide us with clues to the key driving forces.

We are also unaware of research focusing on the magnitude of the differences in adult outcomes by the socioeconomic composition of schools or postcodes, both of which may be important and emerging sources of inequalities. This motivated us to investigate the role of childhood income at different levels of aggregation on adult outcomes in these cohorts.

We focused on five dichotomous adult outcomes, each of which measured different dimensions of socioeconomic status and demographic processes: premature mortality, education, unemployment, partnership status and having children. Previous research has mainly used single outcome variables, such as education or adult income, in their analyses. Adulthood socioeconomic and demographic circumstances, however, can hardly be measured with a single outcome variable. Our focus on many outcomes simultaneously, also known as an outcome-wide analysis [28], thus provides a more nuanced picture of the trends in inequalities by childhood socioeconomic environment.

## Materials and methods

We used the FOLK full-population register datasets. The datasets are constructed, curated and maintained by Statistics Finland. The metadata are openly available online [29,30]. The FOLK datasets are constructed by using multiple administrative registers and combining them by using personal identification numbers. The FOLK datasets consist of several modules, each of which has been approved by the Ethical Committee of Statistics Finland. In this study we combined four modules: the FOLK basic module for the years 1996–2019, consisting of data on parental income and on the five outcomes (TK/1185/07.03.00/2021); the FOLK child-parent-year linking dataset, consisting of data on parental identification numbers (TK/1185/07.03.00/2021); EDUC TYHR dataset, consisting of data on upper primary school identification number (TK/905/07.03.00/2023); and a tailored module on personal area postcode numbers (TK/1185/07.03.00/2021). We focused on the complete Finnish population born between the years 1981 and 1989, identified in the FOLK basic module (the starting population n = 584,564). The data was analysed in the period between 1. May 2023 and 29. February 2024. The data was pseudonymised and researchers are not allowed to identify individuals.

For each cohort, we measured income aggregated at parental, school and postcode levels in the calendar year when the cohort members turned 15 years of age. Important decisions on educational pathways between general upper secondary school and vocational education must be made at this age. The income measure used in this study was the standard net yearly income constructed by Statistics Finland. Net income is the sum of salary, capital income, income from

enterprises and transfers minus taxes and tax-deductible expenses. Negative income values were set to zero by Statistics Finland. To preserve privacy, Statistics Finland round income figures to hundreds of euros. The income in the highest annual income percentile was replaced by the median value of this group.

For the parental aggregated income, we calculated the sum of the parents' disposable income and, if the person had two parents in our data (either biological or social parents), divided this by two. We were unable to identify parents for 10,242 individuals (1.8 percent of the starting population). We thus obtain an income per parent, which is not equivalent to a household income measure. Our data limits us from considering more complex family structures or household size. The family income-based analysis included 569,483 individuals for mortality analysis (for other outcomes the sample size was smaller due to missing data, shown in supplementary file 1)). For the school aggregated income, we used this parental income measure and calculated the median parental income of the cohort members in each school using the school identification number. The school identification number was not found for 10,854 individuals in the starting population (1.9 percent). We used the median because it is not affected by extreme values. We excluded schools with less than five cohort members. The school-based measures included 569,114 observations. For the postcode aggregated income, we calculated the median income among all adult residents in the given postcode, not just the cohort members' parents. Postcodes was not found for 5,616 persons (1 percent). We excluded postcodes with less than 50 adult residents. The postcode based measured included 578,331 observations.

These aggregated measures were calculated separately for each cohort. Finally, we divided the cohort members into tenths by the three income measures – parental, school, and postcode – and cohorts. The group sizes fluctuated slightly in school and postcode income tenths due to the varying sizes of schools and postcodes.

People with a recorded death date between the calendar years they turned 15 and 30 were coded as deceased. Data on mortality were originally derived from the records of the Digital and Population Data Service Agency. Being partnered was defined as having a married or cohabiting family status (people registered as children in their household were considered as not partnered). Having children was coded from a family status variable and those with the following family statuses were regarded as having children in their families: a married couple with children, a mother with children, a father with children, a cohabiting couple with common children, or a cohabiting couple with non-common children. Children here can be above 18 years of age and not necessarily biological children. Those in the position of being a child in a family were considered as not having children. Low education was coded from an educational qualification variable and defined as having no upper-secondary nor higher qualifications. The educational data are originally derived from the Finnish National Agency for Education. The long-term unemployment outcome was coded from a variable of the unemployment days in a given year. People with six or more unemployment months (more than 165 days) in the calendar year were coded as long-term unemployed. These five adult outcomes were taken from the FOLK basic module, specifically, the calendar year when the cohort members turned 30 years of age. In terms of the mortality data, we assume no mortality among the population who emigrated. The rest of the outcome measures were set as missing for people who were deceased or had emigrated. This study does not include persons who moved to the country after their 15th birthday.

We performed a set of descriptive analyses. We first calculated the conditional means of the outcomes by cohort and income measure. We present these results in plots. We also calculated the risk differences and risk ratios while using the 10th income tenth as a reference group. To illustrate variability due to number of cases, we provide 95 percent confidence intervals for the proportions (estimated via Wilson method), risk ratios (estimated via Poisson regression models with robust standard errors) and risk differences (estimated via linear probability regression models with robust standard errors).

To assess the trends we then calculated absolute and relative inequality measures [31]. The slope index of inequality (SII) summarised the differences in absolute terms. SII assumes a monotonic and linear relationship between the income tenths and the outcomes. SII can be interpreted as the risk difference between the first and last tenths while assuming a monotonic linear relationship between the socioeconomic variable and the risk of the outcome. We used linear regression

models to calculate the SII, in which we regressed the conditional mean of the outcome on the income measure treated as a continuous variable while weighting the model by the population size in each decile. The relative inequality index (RII) summarises the relative differences. The RII assumes monotonic multiplicative relationships between income tenths and the risk of the outcomes. The RII is interpreted as the risk ratio between the first and last tenths while assuming a similar trend across all income tenths. We used the predictions from the linear model described above and divided the predicted value of the highest group and the predicted value of the lowest group to obtain an RII value.

## Results

Table 1 shows the parental, school and area aggregated income tenths and the conditional means of the five outcomes. We provide the number of people in each group and the mean income at each level of aggregated income in the supplementary data file. The range between the 1st and the 99th percentile of income in each income decile is provided in S1–S3 Tables. For example, the range for family income was 2,750–13,800 euros for the first and 33,500–80,600 euros for the last decile in the cohort of 1989.

Fig 1 shows the share of deceased. Some 0.9 percent of the cohorts died between the calendar years of their 15th and 30th birthdays, with a lower share in the more recent cohorts. The share of deceased was consistently larger among the cohort members with lower parental and school income while the mortality decline across cohorts was more significant for those with high-income parents and who attended more high-income schools. The risk ratio between the lowest and highest parental income tenths was, on average, 1.8 and ranged from 1.1 (95% confidence interval 0.8–1.6) in the cohort of 1983 in the cohort of to 2.5 (1.7–3.8) in the cohort of 1986. The corresponding risk difference was 0.4 percentage points, ranging from 0.1 (−0.2–0.4) in the cohort of 1983 to 0.8 (0.45–1.1) in the cohort of 1986. An association of a similar magnitude was observed for school income. The lowest school income tenths had, on average, a 1.7-fold risk of death compared to the highest with this figure ranging from 1.1 to 2.8 depending on the cohort. The association between postcode income and mortality was, however, less consistent across the cohorts.

Fig 2 shows the proportion with low education. The proportion of the cohort members with low education decreased from 11 percent in the cohort of 1981–9 percent in the cohort of 1989, conditional on being alive. In all cohorts, parental income was strongly associated with low education. The risk ratio between the first and the last tenths was, on average 3, ranging from 2.6 (2.3–2.9) in the cohort of 1981 to 3.4 (3–3.9) in the cohort of 1987. The mean risk difference was 10.1 percentage points, ranging from 9.5 (8.4–10.6) in the cohort of 1981 to 11.2 (10.1–12.3) in the cohort of 1984. The school-aggregated income had a weaker link (an average risk ratio of 1.3) to low education. Postcode income had a weak

**Table 1. The number of observations and cases by birth cohorts.**

|  | Children | | Deceased | | LT unemployment | | No education | | Partnership | |
|---|---|---|---|---|---|---|---|---|---|---|
| h Cohort | n | cases | n | cases | n | cases | n | cases | n | cases |
| 1981 | 61367 | 26357 | 64414 | 659 | 62263 | 3859 | 62263 | 7123 | 61367 | 40415 |
| 1982 | 63893 | 27081 | 66956 | 649 | 64754 | 4303 | 64754 | 7413 | 63893 | 41667 |
| 1983 | 64433 | 27376 | 67502 | 648 | 65307 | 5158 | 65307 | 7473 | 64433 | 41734 |
| 1984 | 62845 | 26127 | 65834 | 607 | 63745 | 5641 | 63745 | 6901 | 62845 | 40069 |
| 1985 | 60610 | 24654 | 63487 | 554 | 61453 | 6128 | 61453 | 6296 | 60610 | 38268 |
| 1986 | 58442 | 23135 | 61522 | 571 | 59341 | 5796 | 59341 | 6160 | 58442 | 36502 |
| 1987 | 57612 | 21683 | 60658 | 530 | 58520 | 4793 | 58520 | 5609 | 57612 | 35263 |
| 1988 | 60774 | 22010 | 63995 | 549 | 61734 | 4250 | 61734 | 5762 | 60774 | 36674 |
| 1989 | 60794 | 20910 | 63963 | 522 | 61729 | 3895 | 61729 | 5543 | 60794 | 36184 |
| Total | 550770 | 219333 | 578331 | 5289 | 558846 | 43823 | 558846 | 58280 | 550770 | 346776 |

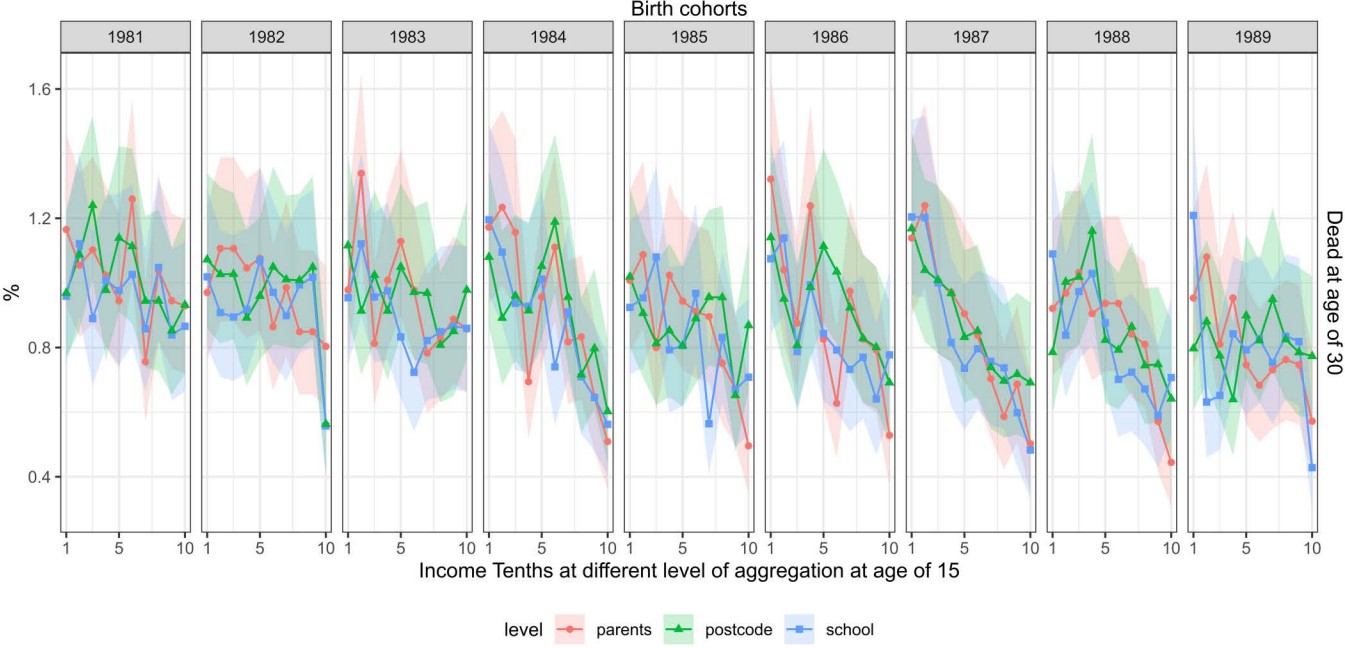

**Fig 1. The share of deceased, measured at the age of 30 (rows) and childhood income at different levels of aggregation at age of 15 (separate lines) across the cohorts of 1981–1989 (columns).** FOLK dataset. Number of observations and mean parental income in each cell shown in supplementary material file.

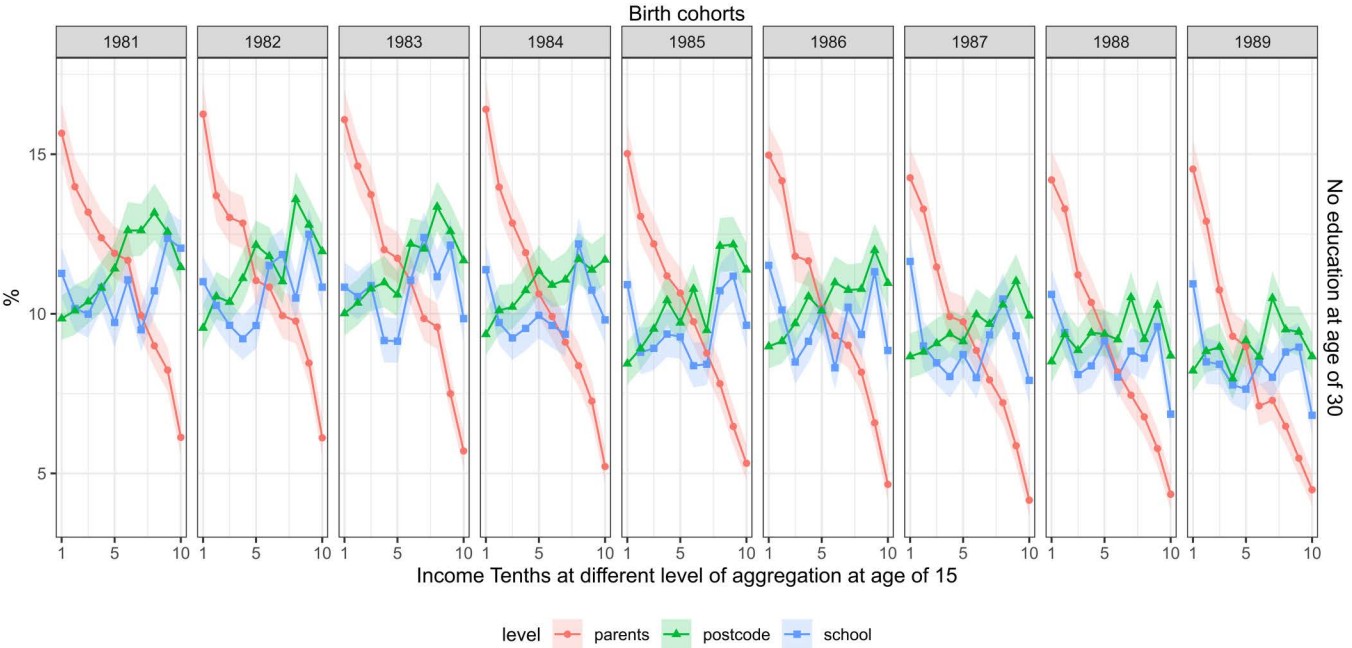

**Fig 2. The share with no education, measured at the age of 30 (rows) and childhood income at different levels of aggregation at age of 15 (separate lines) across the cohorts of 1981–1989 (columns).** FOLK dataset. Number of observations and mean parental income in each cell shown in supplementary material file.

inverse association with educational outcomes (an average risk ratio of 0.9). This association was insignificant in many of the cohorts.

Fig 3 shows the proportion of long-term unemployed in the parental, school and postcode aggregated income tenths. The rate of long-term unemployment varied significantly across the cohorts, peaking for the cohort of 1985 when some 10 percent of the cohort had at least six months of unemployment. Unemployment for this cohort was measured in 2015, the year the national unemployment rate was at its highest during our sample period (see S1 Fig). Across the cohorts, higher parental income was linked to a lower rate of long-term unemployment. The lowest parental income group had, on average, a 2.5-fold risk of long-term unemployment compared to the highest parental income group (the corresponding risk difference was 6.5 percentage points.). For example, for the birth cohort of 1989, the lowest parental income tenth had 3.09-fold risk of long-term underemployment compared to the highest income tenth (2.6–3.6). The school income tenths showed a stronger link to long-term unemployment (an average risk ratio of 1.8) than postcode income (an average risk ratio of 1.4).

Fig 4 shows the share of the cohorts who were married or cohabiting at the end of the calendar year of their 30th birthday. This share declined substantially, from 66 percent in the cohort of 1981–60 percent in the cohort of 1989. On average, across the cohorts, parental income did not show a consistent link with being married or cohabiting (an average risk ratio of 0.9), but in younger cohorts, people with a higher parental income were more likely married or cohabiting than those with a lower parental income. In the cohort of 1989, the risk ratio for being in partnership was 0.91 (0.88–0.93) in the lowest vs. highest parental income tenth. School and postcode income were also associated with partnering; on average, people from the lowest tenth of school or postcode income tenth had a 1.1-fold risk of being married or cohabiting compared to their peers from the highest income tenths. However, these associations attenuated in more recent cohorts.

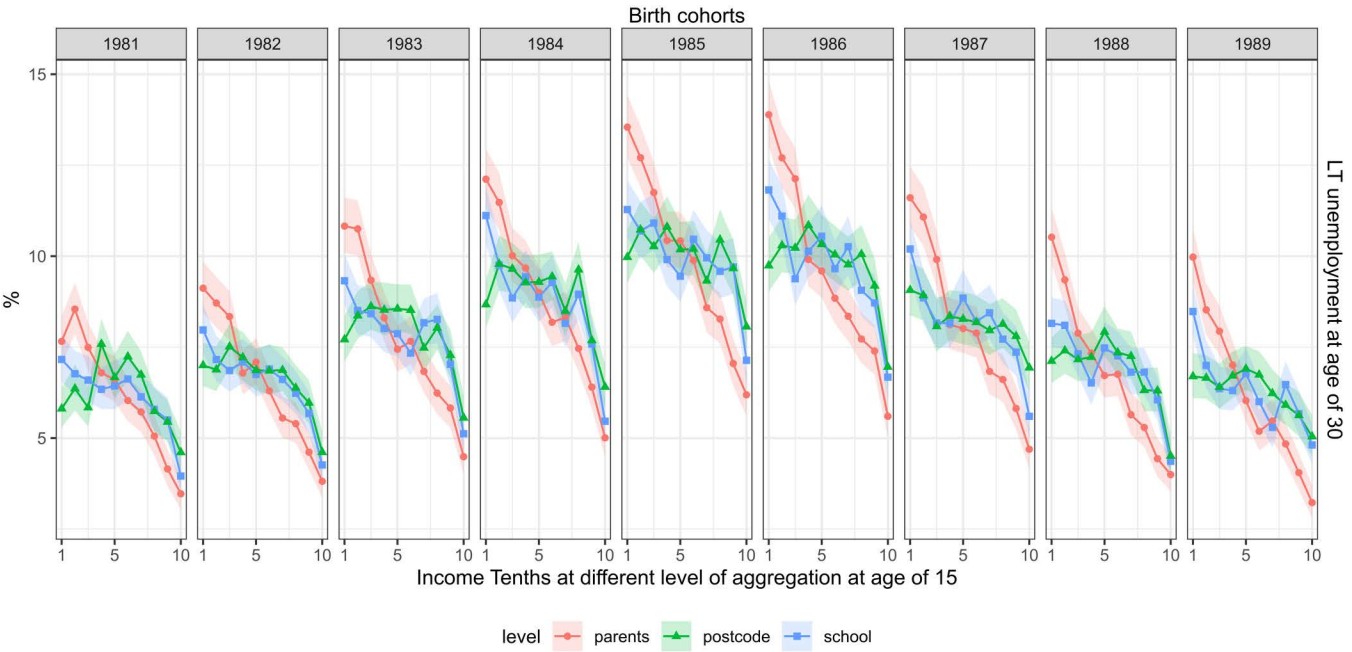

**Fig 3. The share long-term unemployed, measured at the age of 30 (rows) and childhood income at different levels of aggregation at age of 15 (separate lines) across the cohorts of 1981–1989 (columns).** FOLK dataset. Number of observations and mean parental income in each cell shown in supplementary material file.

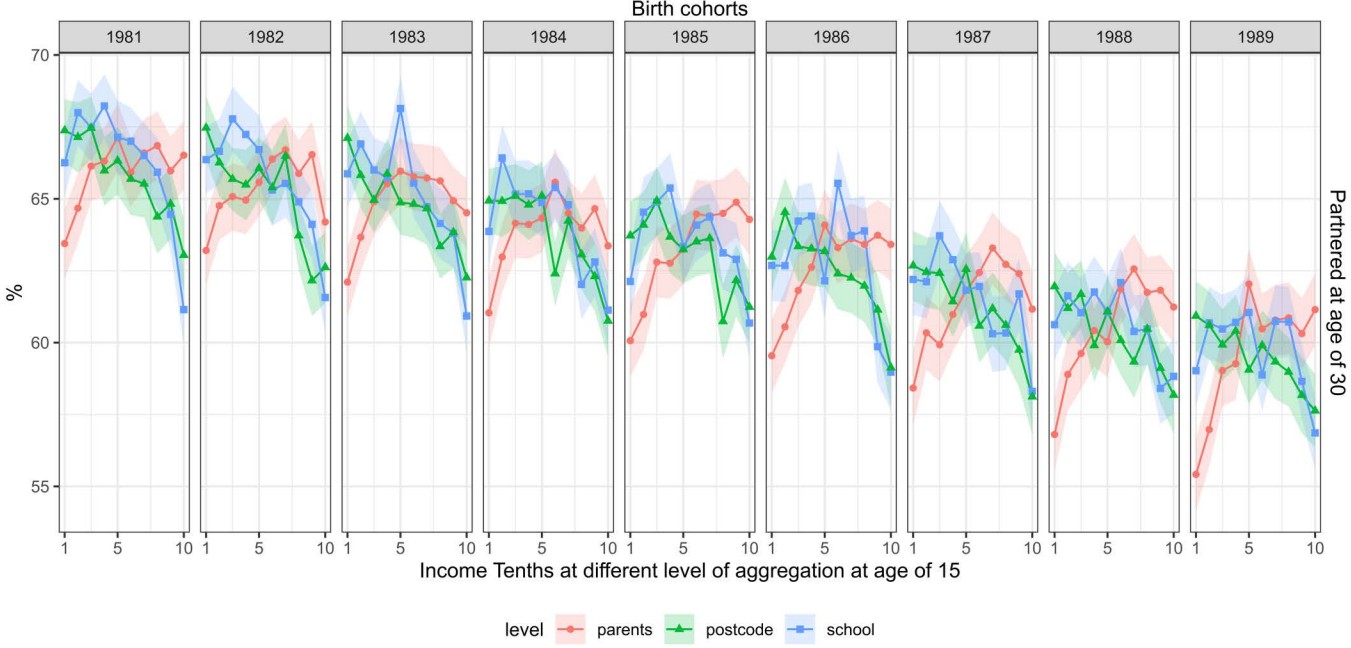

**Fig 4. The share with a partnership, measured at the age of 30 (rows) and childhood income at different levels of aggregation at age of 15 (separate lines) across the cohorts of 1981–1989 (columns).** FOLK dataset. Number of observations and mean parental income in each cell shown in supplementary material file.

Fig 5 shows the association between measures of childhood income and the adult outcome of having children in the family at age 30. The share of the cohort members in families with children declined across the cohorts, from 43 percent in the cohort of 1981–34 percent in the cohort of 1989. Parental, school, and postcode income showed a similar association with this outcome. People in the lowest parental income tenth had, on average, a 1.2-fold risk of being a parent, compared to the highest parental income tenth. This corresponded to a risk difference of 7.6 percentage points. This association remained similar across the cohorts. For example, in the cohort of 1989, the risk ratio was 1.3 (1.2–1.4).

The trends in the absolute inequalities, measured with the SII measure, and relative inequalities, measured with the RII measure, by five outcomes (rows) and levels of aggregation (separate lines), are shown in Fig 6. For mortality, there was an indication of increasing relative inequalities at every level of income aggregation until the cohort of 1987, but these inequalities declined for the cohorts of 1988 and 1989. Similar but smaller trends were observed in absolute terms. The absolute inequalities in low education increased by parental income and to a lesser extent, by school income across the cohorts. Similarly, the relative inequalities increased by parental income and, to a larger extent, by school income. Absolute inequalities in long-term unemployment were strongest in the times of high unemployment (cohorts 1985–1986; cf. Fig A1) but, in contrast, relative inequalities were also lowest during these times. The absolute and relative inequality measures for partnering declined across the cohorts while these measures increased or remained stable for being part of a family with children.

## Discussion

We have examined how differences in income in childhood associate with adult outcomes in nine Finnish cohorts born in the period 1981–1989. We descriptively compared the associations at the three levels of income aggregation – parents, schools and postcodes – and monitored these associations across cohorts. We took an outcome-wide approach by looking simultaneously at five demographic and socioeconomic outcomes, the results of which are discussed below.

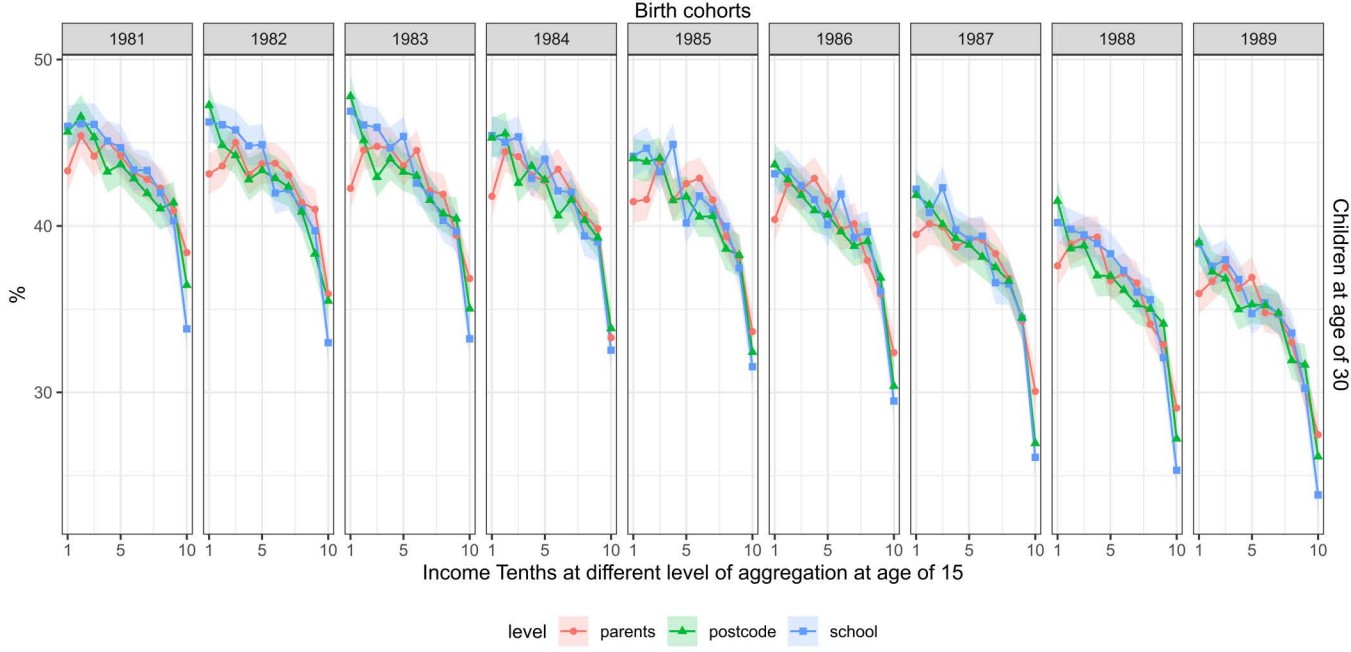

**Fig 5. The share with children, measured at the age of 30 (rows) and childhood income at different levels of aggregation at age of 15 (separate lines) across the cohorts of 1981–1989 (columns).** FOLK dataset. Number of observations and mean parental income in each cell shown in supplementary material file.

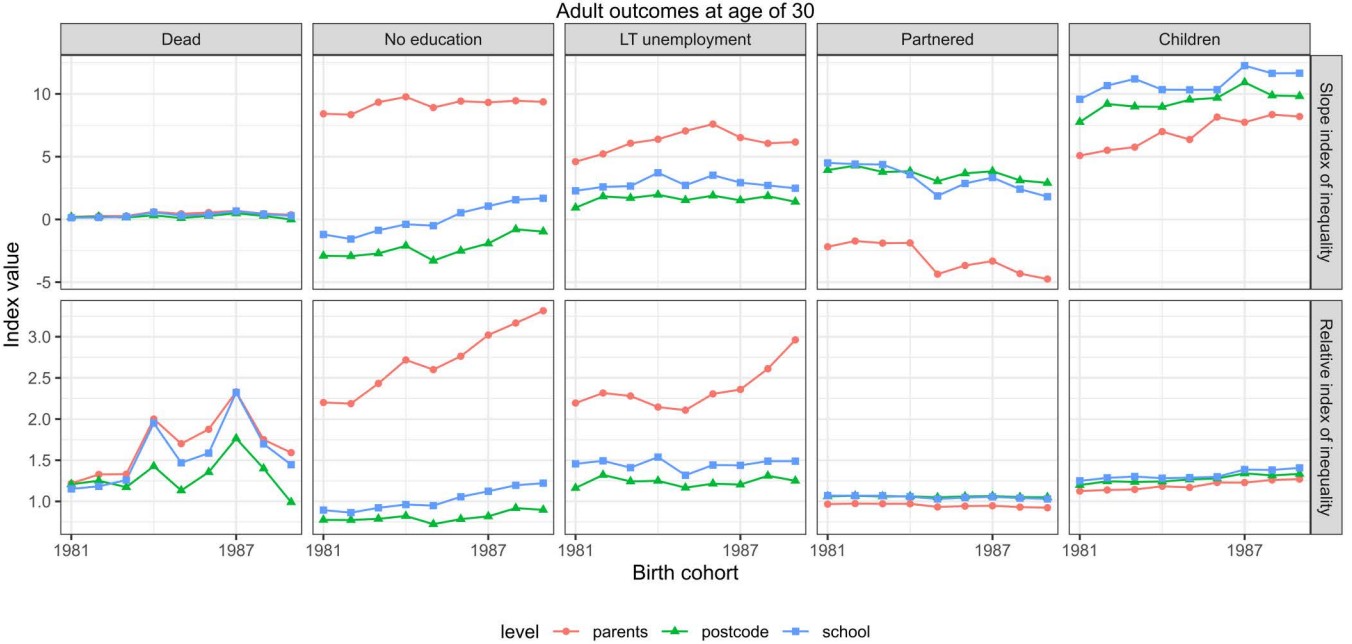

**Fig 6. The trends in absolute inequalities, measured with the SII measure and relative inequalities, measured with the RII measure, by five outcomes (rows) and levels of aggregation (separate lines).** FOLK dataset.

We found a higher risk of premature mortality among those with low childhood income at the parental and school level of aggregation. A large number of studies in the Nordic countries have focused on trends in health inequalities [32,33] but we identified only two looking at mortality in younger ages. Saarela and Finnäs reported that parental social class predicted mortality due to external causes in people aged 15–30 years [34]. Remes et al. found higher parental education-based inequalities in mortality at ages 14–19 than at ages 20–24 [33]. Our findings suggest an increasing association between childhood income and a higher risk of early mortality, and we attribute this finding to the fact that mortality declined more among those with higher childhood income.

We observed substantial differences in the education and employment outcomes by parental income but little by school or postcode income. In turn, school aggregated income had a more consistent link than postcode aggregated income. This may suggest that parental and, to a much lesser extent, school-level resources, or factors associated with these resources, play a key role in adult life trajectories.

Previous Finnish studies do not conclusively report on trends in educational inequalities. Some studies have reported stable absolute inequalities in educational outcomes, such as grade point average [35,36] but others report increasing inequalities. For example, Härkönen and Sirniö report that in the cohorts 1960–1985, the intergenerational transmission of education increased, particularly among women [37]. A handful of these studies focused on area-based inequalities in schooling outcomes [7,38] but such studies have not monitored the trends over time. Our finding supports the claim about the increasing polarisation of low education by childhood socioeconomic environment. Nevertheless, it remains important here to highlight that we see only a small role for school income-based differences in the first place.

Few studies have investigated how childhood socioeconomic status is associated with later unemployment. A previous Finnish study reported that parental unemployment, low education and social assistance use predicted offspring unemployment at the age of 22 [19]. In another Finnish study, parental unemployment was linked to a persistent unemployment trajectory among young people [39]. Our study adds to these by showing that this association was elevated, in absolute terms, in times of high unemployment in 2015.

We investigated how childhood income relates to two family formation outcomes at the age of 30. We observed that the decline in the rates of partnership status coincided with a changing role in respect of childhood income. In earlier cohorts born in 1981, parental income was not linked to partnership status. But the associated with a higher prevalence of cohabitation or marriage emerged in the more recent cohorts because the decline in cohabitation or marriage was more rapid among those with low parental income.

In terms of the having children in the family outcome, we did not observe changes in the role of childhood income. This is despite the major decline in the prevalence of this outcome. We found that low childhood income, at all examined levels, was associated with a higher prevalence of having children in one's family. Previous research has reported that, in Finland and Nordic countries more generally, those with lower educational levels tend to have higher rates of childlessness [40]. A previous Finnish study with population born in the period 1940–1950 reported that lower parental education was associated with a smaller number of children among men [41]. These studies contradict our findings. However, this is explained by our outcome measurement period, 30 years of age, as people with a higher socioeconomic status tend to form families later in life [42]. We did not observe consistent differences at age of 35 by childhood income measures in our supplementary analyses.

We used income as a measure of childhood socioeconomic environment but this measure is not without limitations. First, income does not perfectly present the actual living standards of the households because it does not consider consumption needs or extra costs linked to different conditions. Secondly, due to data constraints, we were not able to take into account household size in order to calculate equivalised household income. Due to these reasons, we replicated all phases of our analyses using wealth instead of income as a measure of childhood socioeconomic environment. We observed some similarities and differences. For example, school aggregated wealth has a stronger link to socioeconomic outcomes than income (our wealth measure was not comparable across the cohorts). These supplementary results are

reported in our shinyapp: https://lopputulemat.shinyapps.io/05_shiny_app/. Nevertheless, subsequent studies are needed with a more detailed measures of childhood socioeconomic circumstances.

This study remains agnostic about the potential drivers of the observed trends. However, one potential driver of the increasing differences by childhood income in some outcomes may be the growing income inequality in the country. During the study period, income inequality increased in Finland according to several measures, including the Gini index (from 0.22 in 1996, when income was measured for the 1981 cohort, to 0.28 in 2004), and the ratio of incomes between the highest and lowest deciles (from 3.1 to 4.1), according to Statistics Finland [43]. These changes are reflected in S1 Table, which shows that the differences in income bands between the first and last deciles increased across the cohorts.

In this study we aimed to describe factual population processes, not causal effects. We do not make causal claims from parental, school or postcode level resources to adult outcomes. We, therefore, are unable to put forward here any policy recommendation on what works in terms of promoting equal opportunities. There is, however, a wealth of causal evidence showing that intervening on childhood resources can have meaningful effects on later outcomes [44]. For example, a recent Norwegian study suggested that to reduce social inequality in educational outcomes, policies should emphasise the need to address the issue of the unequal support available for children with socioeconomic difficulties [45]. In a similar vein, the key policy takeaway stemming from our findings is that more robust policy interventions are required to reduce the link between childhood parental resources and adult outcomes.

## Supporting information

**S1 Fig. The Finnish unemployment rate during the sample period.** The figure plots the unemployment rate the year each of the studied cohorts (born 1981–1989) turned 30. Source: Statistics Finland. https://pxdata.stat.fi/PxWeb/pxweb/fi/StatFin/StatFin__tyti/statfin_tyti_pxt_13ak.px/table/tableViewLayout1/.
(DOCX)

**S1 Table. Family Income 1st and 99th percentile range by Decile and Year.**
(DOCX)

**S2 Table. School Income 1st and 99th percentile range by Decile and Year.**
(DOCX)

**S3 Table. Postcode Income 1st and 99th percentile range by Decile and Year.**
(DOCX)

**S4 Table. All produced estimates.**
(CSV)

## Author contributions

**Conceptualization:** Aapo Hiilamo, Niilo Luotonen, Lauri Mäkinen, Tiina Ristikari.

**Data curation:** Niilo Luotonen.

**Formal analysis:** Niilo Luotonen.

**Investigation:** Aapo Hiilamo, Lauri Mäkinen, Tiina Ristikari.

**Methodology:** Aapo Hiilamo, Niilo Luotonen, Lauri Mäkinen, Tiina Ristikari.

**Software:** Aapo Hiilamo.

**Supervision:** Tiina Ristikari.

**Visualization:** Aapo Hiilamo.

**Writing – original draft:** Aapo Hiilamo.

**Writing – review & editing:** Aapo Hiilamo, Niilo Luotonen, Lauri Mäkinen, Tiina Ristikari.

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
