## [Decision Letter · Decision Letter 0]

Dear Dr. Hiilamo,

Thank you for submitting your manuscript to PLOS ONE. After careful consideration, we feel that it has merit but does not fully meet PLOS ONE’s publication criteria as it currently stands. Therefore, we invite you to submit a revised version of the manuscript that addresses the points raised during the review process.

We look forward to receiving your revised manuscript.

Kind regards,

Sreeram V. Ramagopalan

Academic Editor

PLOS ONE

Reviewers' comments:

Reviewer's Responses to Questions

**Comments to the Author**

1. Is the manuscript technically sound, and do the data support the conclusions?

Reviewer #1: Yes

Reviewer #2: No

2. Has the statistical analysis been performed appropriately and rigorously?

Reviewer #1: Yes

Reviewer #2: I Don't Know

3. Have the authors made all data underlying the findings in their manuscript fully available?

Reviewer #1: Yes

Reviewer #2: Yes

4. Is the manuscript presented in an intelligible fashion and written in standard English?

Reviewer #1: Yes

Reviewer #2: Yes

Reviewer #1: Thank you for your manuscript entitled: Adult outcomes by parental, school and postcode aggregated income in childhood – a descriptive analysis of the cohorts 1981-1989 in Finland.

This is a well design manuscript that has the potential as a candidate for publication.

Reviewer #2: Hiilamo and colleagues use population-based registers to describe temporal changes in the relationship between childhood socioeconomic environment and adult outcomes for the Finnish population born in each calendar year (birth cohorts) during 1981 and 1989. Childhood socioeconomic environment is measured at age 15 based on income at the parental, school and postcode-level, and the study outcomes assessed at age 30 are mortality, educational attainment, long-term employment, relationship status and having had children. Outcomes for each birth cohort are assessed across income deciles (low to high) with risk ratios/differences calculated against the highest (10th) income decile. This is a largely descriptive and unadjusted study, which the authors strongly motivate the need for in the paper’s introduction. The authors report associations between low parental and school income with most outcome measures, with the evidence for parental income the more consistent/compelling, while results for school income are weaker, and postcode income the most inconsistent/harder to interpret. Overall as a reader, it is difficult to assess whether the authors interpretations of the data hold as the result section of the main article is lacking key data.

Major comments

•The main article does not report cohort denominator sizes, numbers of outcome events or the income bands for each decile, and without these it is difficult to fully interpret the data. I strongly suggest these are added in the main text. These are reported to be given in the supplement, however I was unable to locate/review this file.

•There are no confidence intervals reported for the risk ratios/differences and again without these it is difficult to understand the robustness of the findings so also suggest these are added.

•Methods section describes steps taken to prepare the dataset for analysis, e.g cleaning negative values, excluding certain schools/postcodes, handling individuals who emigrated. It would be helpful to quantify how many individuals/observations/record these relate to. Perhaps a data flow chart which details the cohort attrition steps or some other visual could be useful to capture this information succinctly.

•The strength/limitations section of the discussion is a little bit underdeveloped in my opinion. It states that income as measure is not without limitations and provides some sensitivity analyses using an alternative variable (wealth), however doesn’t actually outline what the limitations actually are, so adding these would be informative.

•Some of the language/claims made in the discussions are a little too strong for my liking (e.g. “we found a 80 percent higher risk of premature mortality … at the parental and school level of aggregation”- however the risk ratio reported for schools is 1.7) Suggest careful review/revision to ensure accurate and measured reporting throughout the discussion.

Minor comments

•The authors use the phrase the ‘good life’ on a few occasions e.g. “ What constitutes the ‘good life’…”, however I personally find this concept a little abstract so suggest either providing a definition or consider using alternative wording

•The authors suggest a decline in the number of marriages in Finland “number of new marriages declined from 30 557 in 2011 to 22 296 in 2019 according to Statistics Finland, amounting to a drop of 27 percent (Statistics Finland, 2023b).”, which I don’t dispute. However the 27% drop reported is in terms of absolute numbers and does not consider differences in the underlying population sizes between 2011 and 2019. Suggest these figures would be more accurately reported as a percentage of the population size.

•The method section includes some comments comparing with other studies eg. “ Previous research has mainly used single outcome variables”. Suggest this would be better suited to the introduction or discussion sections and reserve the methods section for details on the present study only.

**Do you want your identity to be public for this peer review?** For information about this choice, including consent withdrawal, please see our Privacy Policy

Reviewer #1: **Yes: ** Behzad Foroutan

Reviewer #2: No

---

## [Author Response · Author response to Decision Letter 1]

21 May 2025

Dear Dr Ramagopalan,

We thank you for the opportunity to revise our manuscript "Adult outcomes by parental, school and postcode aggregated income in childhood – a descriptive analysis of the cohorts 1981-1989 in Finland”

We greatly appreciate the comments provided by the Reviewers. These comments have helped us to improve the manuscript.

We have carefully considered each point. In the attached response letter, we address these comments and describe our responses and the modifications in the manuscript. In the manuscript, these changes are marked with track changes. In addition to the changes highlighted in the response letter, we have corrected some typos in the manuscript.

We hope that the manuscript is now fully acceptable for the publication.

Your sincerely,

Aapo Hiilamo, PhD

Reviewer #1: Thank you for your manuscript entitled: Adult outcomes by parental, school and postcode aggregated income in childhood – a descriptive analysis of the cohorts 1981-1989 in Finland.

This is a well design manuscript that has the potential as a candidate for publication.

Response #1: Thank you for your positive assessment.

Reviewer #2: Hiilamo and colleagues use population-based registers to describe temporal changes in the relationship between childhood socioeconomic environment and adult outcomes for the Finnish population born in each calendar year (birth cohorts) during 1981 and 1989. Childhood socioeconomic environment is measured at age 15 based on income at the parental, school and postcode-level, and the study outcomes assessed at age 30 are mortality, educational attainment, long-term employment, relationship status and having had children. Outcomes for each birth cohort are assessed across income deciles (low to high) with risk ratios/differences calculated against the highest (10th) income decile. This is a largely descriptive and unadjusted study, which the authors strongly motivate the need for in the paper’s introduction. The authors report associations between low parental and school income with most outcome measures, with the evidence for parental income the more consistent/compelling, while results for school income are weaker, and postcode income the most inconsistent/harder to interpret. Overall as a reader, it is difficult to assess whether the authors interpretations of the data hold as the result section of the main article is lacking key data.

Major comments

•The main article does not report cohort denominator sizes, numbers of outcome events or the income bands for each decile, and without these it is difficult to fully interpret the data. I strongly suggest these are added in the main text. These are reported to be given in the supplement, however I was unable to locate/review this file.

Response #2: We sincerely thank the reviewer for this constructive comment. We fully agree and have now added a table to the main text presenting both the denominators and numerators.

Unfortunately, Statistics Finland does not permit the disclosure of income bands, as these bands can theoretically correspond to individual persons. However, we agree that it is important to provide a sense of the income distribution. We therefore have included the 1st and 99th percentiles for each income decile in the supplementary materials.

Given the large of supplementary data, we have developed an interactive Shiny app to allow readers to explore the results in greater detail. Additionally, we provide the aggregated data used to generate the figures in the supplementary materials. We regret that the reviewer was unable to locate this supplementary file. It is now included.

•There are no confidence intervals reported for the risk ratios/differences and again without these it is difficult to understand the robustness of the findings so also suggest these are added.

Response #3: Thank you for this important point. We fully agree with the reviewer and have now added confidence intervals to the figures.

We now state in the text:

“To illustrate variability due to the number of cases, we provide 95% confidence intervals for the proportions (estimated via the Wilson method), risk ratios (estimated via Poisson regression models with robust standard errors), and risk differences (estimated via linear probability regression models with robust standard errors).”

Confidence intervals have also been added to the interactive Shiny app: https://lopputulemat.shinyapps.io/05_shiny_app/

Additionally, to improve readability, we have split the original graph into five separate figures.

•Methods section describes steps taken to prepare the dataset for analysis, e.g cleaning negative values, excluding certain schools/postcodes, handling individuals who emigrated. It would be helpful to quantify how many individuals/observations/record these relate to. Perhaps a data flow chart which details the cohort attrition steps or some other visual could be useful to capture this information succinctly.

Response #4: Thank you for this suggestion. We agree with the reviewer that it is important to clarify how many observations are actually used and affected by each data-processing step. We have added information on the number of observations excluded or modified at each stage of the data preparation process in the main text.

Unfortunately, we are unable to provide a single flow chart of these steps, as the number of observations varies depending on the level of measurement and the specific outcome being analyzed. Nonetheless, we have aimed to present the information clearly in the revised Methods section.

•The strength/limitations section of the discussion is a little bit underdeveloped in my opinion. It states that income as measure is not without limitations and provides some sensitivity analyses using an alternative variable (wealth), however doesn’t actually outline what the limitations actually are, so adding these would be informative.

Response #5: Thank you for this point. We regret that the limitation section was underdeveloped. We now write that “We used income as a measure of childhood socioeconomic environment but this measure is not without limitations. First, income does not perfectly present the actual living standards of the households because it does not consider consumption needs or extra costs linked to different conditions. Secondly, due to data constraints, we were not able to take into account household size in order to calculate equivalised household income. Due to these reasons, we replicated all phases of our analyses using wealth instead of income as a measure of childhood socioeconomic environment.”

•Some of the language/claims made in the discussions are a little too strong for my liking (e.g. “we found a 80 percent higher risk of premature mortality … at the parental and school level of aggregation”- however the risk ratio reported for schools is 1.7) Suggest careful review/revision to ensure accurate and measured reporting throughout the discussion.

Response #6: Thank you for this. We fully agree that careful language is important, particularly when interpreting comes to relative risks when baseline risks are low. We regret the incorrect figure cited in the discussion and have now corrected.

In addition, we have thoroughly reviewed and revised the discussion. We have also added language emphasizing the need for further research to confirm and extend our findings.

Minor comments

•The authors use the phrase the ‘good life’ on a few occasions e.g. “ What constitutes the ‘good life’…”, however I personally find this concept a little abstract so suggest either providing a definition or consider using alternative wording

Response #7: We agree that the phrase “the good life” may come across as abstract or ambiguous in this context. The term was used three times in the manuscript, and we have now revised each instance. We use clearer and more precise language that better reflects the intended meaning in each of these instances.

•The authors suggest a decline in the number of marriages in Finland “number of new marriages declined from 30 557 in 2011 to 22 296 in 2019 according to Statistics Finland, amounting to a drop of 27 percent (Statistics Finland, 2023b).”, which I don’t dispute. However the 27% drop reported is in terms of absolute numbers and does not consider differences in the underlying population sizes between 2011 and 2019. Suggest these figures would be more accurately reported as a percentage of the population size.

Response #8: We agree with the reviewer here. We have calculated crude marriage rate. We now write “For instance, the number of new marriages declined from 71 per 10 000 adults (30 557 new marriages) in 2011 to 50 (22 296 new marriages) in 2019 according to Statistics Finland, amounting to a drop of 30 percent (Statistics Finland, 2023b).”

•The method section includes some comments comparing with other studies eg. “ Previous research has mainly used single outcome variables”. Suggest this would be better suited to the introduction or discussion sections and reserve the methods section for details on the present study only.

Response #9: Thank you for this point. We fully agree with the reviewer and have moved this paragraph.

We genuinely thank the reviewer for these excellent points.

---

## [Decision Letter · Decision Letter 1]

Dear Dr. Hiilamo,

Thank you for submitting your manuscript to PLOS ONE. After careful consideration, we feel that it has merit but does not fully meet PLOS ONE’s publication criteria as it currently stands. Therefore, we invite you to submit a revised version of the manuscript that addresses the points raised during the review process.

We look forward to receiving your revised manuscript.

Kind regards,

Sreeram V. Ramagopalan

Academic Editor

PLOS ONE

Journal Requirements:

Reviewers' comments:

Reviewer's Responses to Questions

**Comments to the Author**

Reviewer #2: (No Response)

2. Is the manuscript technically sound, and do the data support the conclusions?

Reviewer #2: Yes

3. Has the statistical analysis been performed appropriately and rigorously?

Reviewer #2: Yes

4. Have the authors made all data underlying the findings in their manuscript fully available?

Reviewer #2: Yes

5. Is the manuscript presented in an intelligible fashion and written in standard English?

Reviewer #2: Yes

Reviewer #2: Thank you for inviting my review of the revised manuscript by Hiilamo and colleagues on childhood socioeconomic environment and health outcomes in Finland – the manuscript is strengthened with the edits made by the researchers.

The inclusion of information on cohort denominators, numbers of outcome events and data flow shows this is a reasonably large study with few exclusions due to missing information, which is encouraging. Further, the addition of confidence intervals helps with interpreting the results and examining strength of the conclusions made.

I have just one remaining comment, relating to the information on the income bands in each decile which has been added. This tabulation very interestingly shows there is variation over birth cohorts in the income ranges across deciles i.e. what constituted income decile 1 in 1986 (range 1650 - 10050), differs to decile 1 in 1989 (range 2750 - 13800), similarly decile 10 in 1986 (range 22400 - 43100) versus decile 10 in 1989 (range 33500 - 80600). It would appear to me that income has broadly increased over time, and the gap between the decile 1 and 10 has widened. This sounds entirely plausible, however I would ask the authors to consider whether the underlying changes in the distribution of income over time could potentially influence the result in anyway i.e. if it may have exaggerated any of the risk differences or if it is unlikely to have made a material difference.

Other Minor - Table 1. Perhaps add a final row in the table with the total

**Do you want your identity to be public for this peer review?** For information about this choice, including consent withdrawal, please see our Privacy Policy

Reviewer #2: No

---

## [Author Response · Author response to Decision Letter 2]

9 Jun 2025

Dear Dr Ramagopalan,

We thank you for the opportunity to revise again our manuscript "Adult outcomes by parental, school and postcode aggregated income in childhood – a descriptive analysis of the cohorts 1981-1989 in Finland”

We greatly appreciate the comments provided by the Reviewer. These comments have helped us to improve the manuscript.

We have carefully considered each point. In the attached response letter, we address these comments and describe our responses and the modifications in the manuscript. In the manuscript, these changes are marked with track changes.

We hope that the manuscript is now fully acceptable for the publication.

Your sincerely,

Aapo Hiilamo, PhD

Reviewer #2: Thank you for inviting my review of the revised manuscript by Hiilamo and colleagues on childhood socioeconomic environment and health outcomes in Finland – the manuscript is strengthened with the edits made by the researchers.

The inclusion of information on cohort denominators, numbers of outcome events and data flow shows this is a reasonably large study with few exclusions due to missing information, which is encouraging. Further, the addition of confidence intervals helps with interpreting the results and examining strength of the conclusions made.

I have just one remaining comment, relating to the information on the income bands in each decile which has been added. This tabulation very interestingly shows there is variation over birth cohorts in the income ranges across deciles i.e. what constituted income decile 1 in 1986 (range 1650 - 10050), differs to decile 1 in 1989 (range 2750 - 13800), similarly decile 10 in 1986 (range 22400 - 43100) versus decile 10 in 1989 (range 33500 - 80600). It would appear to me that income has broadly increased over time, and the gap between the decile 1 and 10 has widened. This sounds entirely plausible, however I would ask the authors to consider whether the underlying changes in the distribution of income over time could potentially influence the result in anyway i.e. if it may have exaggerated any of the risk differences or if it is unlikely to have made a material difference.

Response #1:

Thank you for your assessment and this excellent point. Indeed, during the study period income inequality increased in Finland by several measures including the GINI index (from 0.22 in 1996 to 0.28 in 2004 and the ratio of incomes of the top and last deciles of income distribution (3.1 to 4.1) according to the Statistics of Finland. As the reviewer correctly points out these are reflected in our supplementary table. We agree that this increase in income inequality may have strengthened some of the observed trends.

We now write this in the discussion section:

“This study remains agnostic about the potential drivers of the observed trends. However, one potential driver of the increasing differences by childhood income in some outcomes may be the growing income inequality in the country. During the study period, income inequality increased in Finland according to several measures, including the Gini index (from 0.22 in 1996, when income was measured for the 1981 cohort, to 0.28 in 2004), and the ratio of incomes between the highest and lowest deciles (from 3.1 to 4.1), according to Statistics Finland [42]. These changes are reflected in Supplementary Table 1, which shows that the differences in income bands between the first and last deciles increased across the cohorts.”

Other Minor - Table 1. Perhaps add a final row in the table with the total

Response #2:

Thank you! We agree with the reviewer and have added the total row.

---

## [Editor Report · Decision Letter 2]

Adult outcomes by parental, school and postcode aggregated income in childhood – a descriptive analysis of the cohorts 1981-1989 in Finland

PONE-D-24-45407R2

Dear Dr. Hiilamo,

We’re pleased to inform you that your manuscript has been judged scientifically suitable for publication and will be formally accepted for publication once it meets all outstanding technical requirements.

Kind regards,

Sreeram V. Ramagopalan

Academic Editor

PLOS ONE
---

## [Editor Report · Acceptance letter]

PONE-D-24-45407R2

PLOS ONE

Dear Dr. Hiilamo,

I'm pleased to inform you that your manuscript has been deemed suitable for publication in PLOS ONE. Congratulations! Your manuscript is now being handed over to our production team.

Kind regards,

on behalf of

Dr. Sreeram V. Ramagopalan

Academic Editor

PLOS ONE